# A COLLABORATIVE ATTENTION ADAPTIVE NETWORK FOR FINANCIAL MARKET FORECASTING

## ABSTRACT

Forecasting the financial market with social media data and real market prices is a valuable issue for market participants, which helps traders make more appropriate trading decisions. However, taking into account the differences of different data types, how to use a fusion method adapted to financial data to fuse real market prices and tweets from social media, so that the prediction model can fully integrate different types of data remains a challenging problem. To address these problems, we propose a collaborative attention adaptive Transformer approach to financial market forecasting (CAFF), including parallel extraction of tweets and price features, parameter-level fusion and a joint feature processing module, that can successfully deeply fuse tweets and real prices in view of the fusion method. Extensive experimentation is performed on tweets and historical price of stock market, our method can achieve a better accuracy compared with the state-of-the-art methods on two evaluation metrics. Moreover, tweets play a relatively more critical role in the CAFF framework. Additional stock trading simulations show that an actual trading strategy based on our proposed model can increase profits; thus, the model has practical application value.

## 1 INTRODUCTION

In recent years, the use of social media information to predict the financial market has attracted the attention of more and more researchers, and some satisfactory experimental results have been achieved. This is due to the fact that social media information contains investor-related attitudes and subjective sentiments to the financial market, resulting in many investment banks and hedge funds trying to dig out valuable information from media information to help better predict financial markets, which play a key role in predicting the market. At the same time, the efficient market hypothesis (EMH) introduced by Fama (1970) points out that the current price of the asset reflects all the prior information that is immediately available. Therefore, using social media information and the actual price of the current financial market seems to be able to complete the market forecasting more accurately.

To the best of our knowledge, the different types of data fusion methods commonly used in the field of financial market forecasting are mainly concatenation or weighted sums. They are simple, direct, and almost no parameter exchange methods to achieve data fusion (Xu & Cohen, 2018; Perez-Rua et al., 2019; Xu et al., 2019). In the field of computer vision (CV), multimodal fusion methods are mainly used for the fusion of data based on text and images. By realizing the interaction of different information, extracting rich feature information, and obtaining satisfactory experimental results (Baltrusaitis et al., 2019). It can be seen that the multimodal fusion method in the CV field seems to be more conducive to feature exchange and feature supplementation than the current method in the financial field (Lu et al., 2018; Kim et al., 2017). Therefore, we can infer that choosing an adaptive financial data fusion algorithm to achieve the integration of social media information and actual market prices is effective for improving the forecasting effect of the financial market. However, the current financial market forecasting field basically does not consider too many modal fusion methods, which may be one of the reasons that limit the current forecasting effect. In addition, the role of text and historical prices in the fusion method is different from multimodal fusion tasks, such as face recognition, medical aided diagnosis, and visual analysis. The fundamental reason is that the application scenarios are inconsistent, the source data for other tasks is objective, and the financial data is affected by the sentiment of market investors. Taking into account the difference between

tweets and prices, we should pay attention to the effects of both when choosing a fusion method, and use the quantitative analysis of experimental results.

To effectively solve the above problems, we propose a collaborative attention (co-attention) Transformer approach adaptive to financial market forecasting called CAFF, partially inspired by the recent proposed multimodal fusion in bidirectional encoder representation from Transformers (BERT), which was originally developed to utilize Transformer to realize the feature interaction between text and image data (Lu et al., 2019). To verify the rationality of the CAFF method, we included the existing traditional multimodal fusion methods in the experiments for comparison.

In summary, the contributions of our work are as follows:

- To the best of our knowledge, this work is among the first to introduce a co-attention Transformer fusion approach adaptive to financial market forecasting, which is inspired by the multimodal fusion method and takes into account the characteristics of financial data.

- We propose a novel financial market forecasting framework based on the idea of deep fusion to model multisource data analysis. The components of the framework work together to extract tweets and price features in parallel, fuse the features and realize accurate prediction.

- Experimental results on stock market forecasting tasks demonstrate that our proposed method achieves a substantial gain over state-of-the-art methods. Moreover, the results also reveal that under the CAFF fusion framework, the quantitative analysis results show that social media information has played a relatively more critical role.

## 2  RELATED WORK

Some researchers have adopted the method of fusing real market prices and textual information reflecting investor sentiment from social media. In this section, we review the work related to financial market forecasting that fuse prices and text.

A variety of analysis approaches based on text and prices to market forecasting have been proposed in the research literature. For example, Xu & Cohen (2018) proposed a new depth generation model for the stock market forecasting task, which combines text and price signals as the source information. Similarly, Li et al. (2020) also used the concatenation method to integrate the sentiment information contained in the news data and stock prices to predict the Hong Kong stock market. In addition, Xu et al. (2020) proposed a stock movement prediction network based on tweet and stock prices by means of incorporative attention mechanism that combines local and context attention mechanisms through incorporative attention to clean up context embedding using local semantics. The approach also makes use of concatenation. It can be seen that the common fusion methods in the field of financial market prediction are still surface feature fusion without parameter interaction. At present, the mainstream fusion technology in the field of CV has attracted our attention, such as bilinear pooling, attention-based fusion, in which bilinear pooling creates a joint representation space by calculating the outer product to facilitate the multiplicative interaction between all elements in the two vectors, which is obviously not suitable for describing text and representing the value of price changes. Lu et al. (2019) proposed the visual language BERT (ViLBERT) model, which extends the Transformer to the fusion operation of processing images and texts. These extracted features can interact through a parallel attention layer, which provides a creative way for feature optimization, but we have to consider the differences between financial data and CV, speech task and other fields. Therefore, developing a method suitable for financial data with the help of the existing fusion methods is a problem worthy of discussion.

In summary, using an improved deep learning framework based on a fusion algorithm to analyze financial data to achieve financial market forecasting appears to be effective, which can be confirmed via quantitative analysis. On the basis of the above advanced theoretical analysis, our research motivations are as follows. First, we believe that a framework that processes text and prices in parallel and fully fuses them would perform better than a single data source, where the fusion algorithm plays a key role. However, referring to the existing fusion methods in the CV field, the most natural and effective way to process prices may be to consider the original form as 1-D data rather than as a 2-D matrix image Gupta et al. (2020). Moreover, the role of text used to describe the movement of

financial market is inconsistent with prices, which are different from the text and image in the cross-modal task of CV fields. The above problems require an applicable prediction framework based on the fusion method to realize the feature extraction and fusion of tweets and prices with temporality, as well as a matching activation function. Hence, we propose a new framework called CAFF, which uses the Transformer model adapted to financial data and a variety of attention mechanisms to collaboratively predict the movement on the target trading day.

## 3 MODEL OVERVIEW

We propose CAFF, which conforms to the characteristics of financial data, and the complete framework is shown in Figure 1. The upper of the model deeply extracts text features to learn the text representation of tweets, and the lowers extracts market price features in parallel. The operation with Transformer as the main structure in the middle is used to fuse tweets and price information, and the right side is used to process the merged features and realize the prediction of market trends on the target trading day.

### 3.1 THE PROPOSED MODEL: CAFF

Considering that there is a certain lag in the target trading day $d$, we are allowed to simulate and predict other trading days close to $d$ in principle. We can not only predict the trend of the target trading day itself, but also the trend of other trading days during the lag period. For example, if we choose 08/06/2021 as the target trading day, then 03/07/2021 and 07/06/2021 represent the endpoints of the lag period (the lag period is usually 5 days); thus, we capture the relationships between the predictions mainly within this sample range. However, considering the actual conditions of the financial market, we neglected nontrading days in the calculation process to achieve the effect of effectively organizing and utilizing input data. In general, we can predict a series of movements $z = [z_1, z_2, \ldots, z_T]$, where the target trading day is $z_T$ and the rest are auxiliaries.

#### 3.1.1 INPUT REPRESENTATION

Next, we introduce the processing of text and stock prices to obtain the input representation. Text data is regarded as a feature extraction task in NLP domain. Considering the inherent correlation and volatility, we adopt a relatively direct method to deal with stock prices to maintain the original structure.

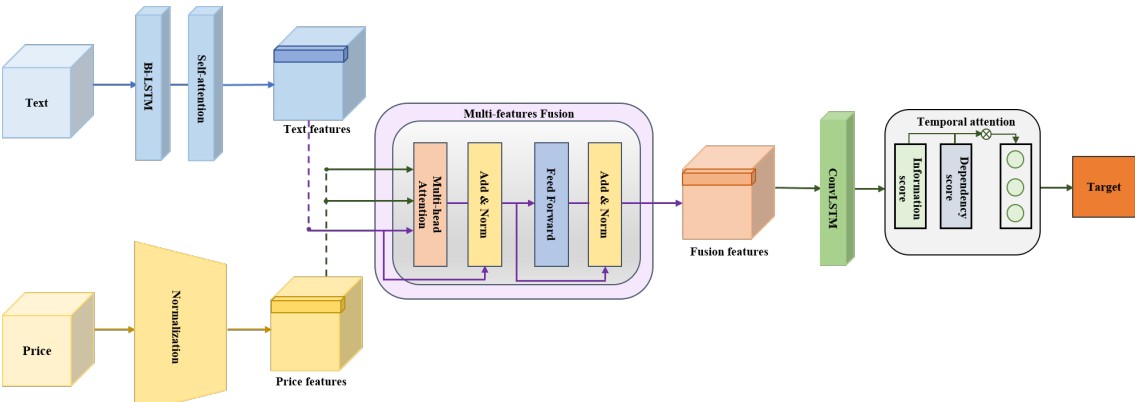

Figure 1: The overall architecture of the proposed stock movement prediction model CAFF.

**Tweet-level Model:** Generally, tweets posted on social platforms for a given stock often contain multiple inconsistent items on a given trading day. To learn more profitable and valuable information from a large number of investor tweets, we adopt the Bi-LSTM and self-attention to obtain the representation as the input of the fusion component. For each sentence $t_i$, we adopt the pre-trained word embedding (GloVe) to project each word tag onto the $d - dimensional$ space and the

pretrained word embedding as the input of Bi-LSTM.

$$f_t = \sigma(W_f \cdot [h_{t-1}, x_t] + b_f) \tag{1}$$

$$i_t = \sigma(W_i \cdot [h_{t-1}, x_t] + b_i) \tag{2}$$

$$\widetilde{C}_t = tanh(W_C \cdot [h_{t-1}, x_t] + b_C) \tag{3}$$

$$C_t = f_t * C_{t-1} + i_t * \widetilde{C}_t \tag{4}$$

$$O_t = \sigma(W_O \cdot [h_{t-1}, x_t] + b_O) \tag{5}$$

$$h_t = O_t * tanh(C_t) \tag{6}$$

$$h = [\overrightarrow{h_t}, \overleftarrow{h_t}] \tag{7}$$

In accordance with the nature of time series data, we use an RNN with Bi-LSTM units and a self-attention mechanism to recursively extract features. The design characteristics of LSTM allow it to model text data (e.g., tweet data, news data) to extract text features and capture the context relationship. Bi-LSTM adds a process of forward and backward concatenation of hidden vectors on the basis of the realization of LSTM. Formulas (1–7) describe the details of Bi-LSTM.

The essence of the attention function can be described as a query (q) to a series of key-value (k-v) mappings. In current research, the *k* and *v* are often the same, that is, $key = value$. Formulas (8–10) describe the details of the attention operations. The only difference between the self-attention mechanism and the abovementioned conventional attention mechanism is that $q = k$. Self-attention mechanisms have become a hot topic in recent research and have been explored in different tasks. The main reason we choose the self-attention mechanism after Bi-LSTM is that the self-attention is calculated for each word and all words, so regardless of the distance between them, long-distance dependency relationships can be captured to achieve the contextual interaction of tweets to obtain key text features.

$$\alpha_i = softmax(F(key_i, q)) \tag{8}$$

$$att((K, V), q) = \sigma_{i=1}^{N} \alpha_i X_i \tag{9}$$

$$attention((K, V), Q) = att((K, V), q_1) \oplus \cdots \oplus \\ att((K, V), q_M) \tag{10}$$

**Price-level Model:** Stock prices more intuitively reflect the real market conditions, while tweets contain objective investor attitudes, which allows the two to complement each other. However, the movement of stock has random volatility and is determined by continuous changes in prices rather than the absolute values of the opening and closing prices. Thus, the original stock price vector of trading day $d$ is not fed directly into the neural network; we employ a normalization strategy to obtain an adjusted closing price. The price adjustment formula is shown below:

$$p_d = [p_d^c, p_d^h, p_d^l] \tag{11}$$

$$p_a = \frac{p_d}{p_{d-1}^c} - 1 \tag{12}$$

where $p_d^c$, $p_d^h$ and $p_d^l$ denote the closing price, highest price and lowest price vectors, respectively.

### 3.1.2 FUSION METHOD BASED ON CO-ATTENTION TRANSFORMER

Due to the excellent performance, an increasing number of Transformer-based frameworks have been proposed to improve various tasks. The input is divided into three parts, which is similar to the self-attention mechanism we mentioned above. In fact, Transformer is composed mainly of the attention mechanisms, where the multihead attention mechanism is the parallel of multiple attention mechanisms, allowing the model to pay attention to information from different representation subspaces at different locations. The follow-up is to input the result of the Add & Norm into the feedforward neural network (FNN) in the fully connected layer.

We improve the fusion method upon this framework to accommodate financial data. Next, we elaborate the complete fusion process in detail. To achieve deep interaction, we first create the $q$, $k$ and $v$ vector; that is, the input representations generated by text embedding and stock price are represented and the Query (Q), Key (K) and Value (V) matrices are generated by multiplying the input representations and the three generated matrices in the training process. Moreover, the conversion process adopts the GELU activation function, which introduces the idea of random regularization. This approach represent a more intuitive and natural understanding of the probability description of the neuron input, which matches a random value of 0 or 1 for the input. Furthermore, some experiments have shown that GELU is better than ReLU and ELU in CV, NLP, and speech tasks. The approximate expression of the GELU is as follows:

$$Gelu(x) = 0.5x(1 + tanh(\sqrt{2/\pi}(x + 0.044715x^3)))$$ (13)

Scaled dot-product attention is the main component of the multihead attention mechanism, so its calculation process is equivalent to reducing the entire implementation. First, we need to calculate the dot product between Q and K to measure the similarity and then divide the value by $\sqrt{d_k}$ to prevent the result from being excessively large, where $d_q$ and $d_k$ are the dimensions of the vectors. Then, the result is normalized to a probability distribution via the softmax function and multiplied by matrix V to obtain the attention vector. The matrix of outputs is as follows:

$$Attention(Q, K, V) = softmax(\frac{QK^T}{\sqrt{d_k}})V$$ (14)

The above calculation process mentions only single-layer attention, while multihead attention is used for parallel calculations. Then, all the values are concatenated to reduce the loss by decreasing the dimensionality. Therefore, we linearly project the queries in different ways with the keys and values $h$ times to learn the linear projections of $d_q$, $d_k$ and $d_v$. Next, we apply the attention function in parallel on each of these queries, keys and values to generate $d_v - dimensional$ output values. The above single values are concatenated in series and projected to form the final required value. By means of the above process, multihead attention allows the model to jointly focus on information from different locations to achieve deep interaction of internal information. The multihead attention function is as follows:

$$MultiHead(Q, K, V) = Concat(head_1, head_2, \ldots, head_h)W^O$$ (15)

where $W^O$ represents the weight matrix. In this work, we employ $h = 5$ parallel attention layers or heads.

Add & Norm denote residual connection and layer normalization, where residual connection has a better effect on deeper neural networks. When the network layer is very deep, the spread of values decreases with the weight. Layer normalization prevents abnormal values due to excessively large or small positions in a certain layer and is applied in training problems when returning the gradient of the neural network to ensure the stability of training. Add & Norm are applied after each subnetwork to make the deep neural network training smoother. Each submodel in the encoder contains a fully connected FNN layer that acts equally on each position. The layer is composed of two linear transformations separated by GELU activation.

$$FFN = max(0, xW^1 + b^1)W^2 + b^2$$ (16)

Although the linear transformation operations used at all positions are the same, exclusive optimal parameters are used between layers.

### 3.1.3 JOINT FEATURE PROCESSING: LSTM

The first step of LSTM is to determine that unnecessary information should be discarded from the cell state, which is realized by the sigmoid layer called the "forgetting door". This layer considers $h_{t-1}$ (previous output) and $x_t$ (current input) and outputs a value between 0 and 1 for each number in the cell state $C_{t-1}$ (previous state). The next step is to determine the information to store in the unit state and to output the information to complete the operation. Formulas (1–6) describe the details of these LSTM operations.

LSTM substantially improves the use of RNN, and the emergence of attention models has led to great progress. Therefore, we further analyze the joint information through temporal attention to achieve the prediction task.

### 3.1.4 JOINT FEATURE PROCESSING: TEMPORAL ATTENTION

We introduce an improved temporal attention mechanism to realize stock prediction on the target trading day and auxiliary trading days. Although different types of temporal attention mechanisms have been proposed, we need a method that is compatible with text and stock price data, that is, a method that accounts for the relationship and key information of both types of data. To this end, as shown in the complete CAFF framework, the temporal attention mechanism is composed of two main parts, namely, the dependency score and information score, which are, respectively, used to obtain the dependency of stock data and more favorable key information.

$$d_t = tanh(W_d[x, h_t] + b_d) \tag{17}$$

where $W_d$ and $b_d$ denote the weight matrix and bias, respectively. Then, the dependency score and information score are obtained by nonlinear projection of $d_t$, and the calculation method is shown in formulas (18–20).

$$v'_i = w_i^T tanh(W_{d,i}D) \tag{18}$$

$$v'_m = d_T^T tanh(W_{d,m}D) \tag{19}$$

$$v = softmax(v'_i \odot v'_m) \tag{20}$$

where $W_{d,i}$ and $W_{d,m}$ are weight parameters and $D$ is the set of all auxiliary trading days. Our goal is to predict the movement on the target trading day, so $d_T$ is used as the final information. To achieve the final prediction, the softmax function is used to obtain the *Up* or *Down* prediction for the target trading day.

$$z_t = softmax(w_y d_t + b_y), t < T \tag{21}$$

$$z_T = softmax(w_T[Zv_T, d_T] + b_T) \tag{22}$$

where $w_y$ and $w_T$ are weight parameters and $b_y$ and $b_T$ are biases. The above calculation steps can achieve a complete prediction of stock market movement on the target trading day.

## 4 EXPERIMENTS AND DISCUSSIONS

### 4.1 DATASET

To verify the actual performance of CAFF and related methods of stock prediction, we adopt stock price and tweets datasets. Stock traders often post personal opinions on social platforms such as Twitter to express their inner practical ideas. On the basis of the heat of discussion, the 88 stocks with the highest capital scale rankings were selected for comparison experiments.

Some stocks show only minimal rates of change, so we implement relevant measures to avoid affecting the experimental results. We chose the upper and lower limits of stock price changes proposed by Hu et al. (2018) for stock trend prediction. Since the stock movement prediction task can be regarded as a binary classification problem, we set two thresholds: $-0.5\%$ and $0.55\%$. For samples with movement ratios of $\leq -0.55\%$ and $>0.55\%$, 0 and 1 were used to represent the fall and rise, respectively, and $38.72\%$ of the target is removed between the two thresholds. According to the two thresholds set above to balance the two predicted categories, 26614 prediction targets are obtained for the entire dataset, of which the two categories account for $49.78\%$ and $50.22\%$, respectively. We divide the dataset into three parts, namely, the training set, test set and validation set, by date.

Table 1: Performance of baseline methods and CAFF in ACC and MCC.

| Model | Accuracy (%) | MCC |
|---|---|---|
| ARIMA (Adebiyi et al., 2014) | 51.39 | -0.0205 |
| TSLDA (Nguyen & Shirai, 2015) | 54.07 | 0.0653 |
| HAN (Hu et al., 2018) | 57.14 | 0.0723 |
| CH-RNN (Wu et al., 2018) | 59.15 | 0.0945 |
| StockNet (Xu & Cohen, 2018) | 58.23 | 0.0807 |
| Adv-LSTM (Feng et al., 2019) | 57.20 | 0.1483 |
| CapTE (Liu et al., 2019) | 64.22 | 0.3481 |
| SMPN (Xu et al., 2020) | 59.74 | 0.1586 |
| CAFF. CON | 60.19 | 0.2436 |
| CAFF. SUM | 59.64 | 0.1589 |
| CAFF. BIL | 58.49 | 0.1377 |
| CAFF. TRA | 57.47 | 0.1163 |
| CAFF (W/O text) | 55.48 | 0.1657 |
| CAFF (W/O price) | 64.89 | 0.3288 |
| CAFF (*lag_size_7*) | 63.77 | 0.3109 |
| CAFF (*lag_size_10*) | 62.50 | 0.2258 |
| **CAFF** | **65.78** | **0.3792** |

## 4.2 TRAINING SETUP

Regarding the specific parameter settings of the experiment, the most basic operation is to obtain the initial word embedding of the text through the GloVe pretraining model, and the word embedding size is 50. The maximum numbers of messages and words in a single message are set to 30 and 40, respectively. The traditional Adam optimizer is more in line with the framework of this paper and is used to train the model with an initial learning rate of 0.001. In addition, the decay rate is set to 0.96, and the decay step is set to 100. We adopt conventional batch operations during the training process, and the batch size is 32. The whole method is implemented using TensorFlow (version 1.14.0).

## 4.3 EVALUATION METRICS

The model predicts the upward or downward movement of stocks for the next day. In this work, we follow some previous research work to evaluate stock prediction (Xie et al., 2013), and standard metrics, including accuracy (ACC) and the matthews correlation coefficient (MCC), are used as performance evaluation indicators. The MCC is essentially the correlation coefficient between the observed and predicted binary classes. As shown in formula (23), *tp*, *fp*, *tn* and *fn* represent the numbers of samples classified as true positives, false positives, true negatives, and false negatives, respectively. The MCC is calculated as follows:

$$MCC = \frac{tp \times tn - fp \times fn}{\sqrt{(tp + fp)(tp + fn)(tn + fp)(tn + fn)}} \tag{23}$$

## 4.4 VERIFYING THE EFFECTIVENESS OF CAFF

### 4.4.1 COMPARISON WITH BASELINE MODELS

To test the diversity and differences of different model architectures, we consider 8 baseline models to compare the performance of the prediction results. The baseline models are designed from the perspective of overall achievement to predict stock movements and are composed of common neural networks, using text or stock prices as input data.

Table 1 shows that the CAFF model achieves the best performance, where numbers in bold font represent the best performance score. An accuracy of 56% is generally considered to be a satisfactory result of binary stock movement prediction (Nguyen & Shirai, 2015). In fact, according to statistical

laws, the probability of binary random prediction is 50%. Although the ARIMA and TSLDA methods in Table 1 have accuracies less than 56%, the statistical models and the deep learning framework based on social media based on these methods are still relatively satisfactory.

In the stage of text feature extraction, both HAN and CH-RNN use a traditional attention mechanism, while CapTE is used in Transformer, which has achieved breakthrough results in NLP and is the main factor leading to the difference in the final experimental results. In addition, CapTE chose the capsule network commonly used in NLP cross-domain direction after processing the text features, which provides researchers with an efficient way to handle temporal text. Adv-LSTM used stock prices as the original analysis data, but the experimental results show that there is room for further improvement. The main reason for this phenomenon may be the range of features covered by the text and price, which may also be affected by the complete framework. The remaining two deep learning frameworks, StockNet and SMPN, both adopt the idea of fusing text and stock prices with the attention mechanism as the main body and adopt the traditional fusion method of concatenation. However, from a quantitative perspective, their results are similar but worse than those of CAFF. Therefore, it is effective for us to use a suitable fusion method and an adaptive deep learning framework for stock price prediction tasks.

### 4.4.2 COMPARISON WITH FUSION METHODS

This subsection provides four solutions that fuse text and stock prices and searches for the best solution to achieve state-of-the-art performance. The four fusion methods are concatenation, weighted summation, bilinear pooling, and Transformer, which changes the input order to quantitatively analyze the fusion method that is most suitable for text and stock data under the current framework, called CAFF. CON, CAFF. SUM, CAFF. BIL, CAFF. TRA respectively.

We can conclude that the fusion operation can be used to quantitatively determine which method is more suitable for CAFF and judge the criticality of the source input data. Concatenation and weighted summation are the most classic algorithms for fusing different modes of information, and many studies have shown that they are effective to a certain extent. Especially in the field of stock market forecasting, concatenation is currently the most commonly used fusion method. According to the quantitative results, CAFF. CON and CAFF. SUM have achieved satisfactory results, even compared with the baseline model in the previous subsection. Bilinear pooling is a feature fusion algorithm for CV proposed in recent years; a large number of researchers continue to improve the method to obtain breakthrough results. However, bilinear pooling does not appear to be suitable for financial data with timeliness. The main reason may be that the extracted features are subjected to the outer product operation during fusion and the dimensionality is forced to be reduced, which causes feature loss. Finally, the Transformer-based fusion algorithm that reverses the input order has the worst performance, further confirming that the text and stock price have different effects on the final forecast results and that the text plays a key role.

### 4.4.3 ABLATION STUDY

To verify the integrity of our CAFF model and make a more detailed analysis of the key components, we construct the following variants in addition to the fully equipped CAFF. CAFF (W/O text) means that the input data do not include the text corpus; that is, only the historical stock prices are considered, and CAFF (W/O price) has a similar meaning. Furthermore, we propose variables with alternative parameter values in which the length of the lag time is modified to 7 days and 10 days; these methods are called CAFF (*lag_size_7*) and CAFF (*lag_size_10*), respectively.

Table 1 shows the performance of CAFF and different variants. CAFF (W/O text) performs the worst, indicating that stock movement is affected by many unfixed factors. Even compared with the baseline model in Table 1, the performance of CAFF (W/O text) is not satisfactory, which proves that historical stock prices play an auxiliary role in the prediction framework. However, CAFF (W/O price) has produced extremely competitive results, which quantitatively confirms the criticality of the text. Therefore, social media contains a lot of valuable market information, and studies have shown that this type of information is useful for predicting stock prices and for other financial tasks (Diaz et al., 2020).

In the ablation study, we additionally modified the lag period of the input data from the current 5 calendar days to 7 days and 10 days. As shown in Table 1, experiments with a lag period of 7

Table 2: Profits comparison between CAFF and CapTE.

| Stock | CAFF | Return(%) | CapTE | Return(%) | Stock | CAFF | Return(%) | CapTE | Return(%) |
|---|---|---|---|---|---|---|---|---|---|
| AAPL | $1287 | 12.87 | $1196 | 11.96 | GOOG | $1003 | 10.03 | $864 | 8.64 |
| ABBV | $910 | 9.1 | $729 | 7.29 | INTC | $1847 | 18.47 | $1200 | 12 |
| BAC | $993 | 9.93 | $941 | 9.41 | ORCL | $985 | 9.85 | $703 | 7.03 |
| CELG | $1567 | 15.67 | $1430 | 14.30 | PFE | $1100 | 11 | $1053 | 10.53 |
| CVX | $1059 | 10.59 | $1005 | 10.05 | WMT | $1572 | 15.72 | $1473 | 14.73 |
| DIS | $892 | 8.92 | $703 | 7.03 | XOM | $1190 | 11.9 | $1092 | 10.92 |

and 10 calendar days did not produce better results than experiments with a lag period of 5. The reasons for this phenomenon are as follows: tweets and historical stock prices are time-sensitive, and the information utilization value is lower farther from the target trading day; moreover, investor comments are more inclined to reflect current market conditions and are not suitable for predicting long-term trading behavior.

## 4.5 MARKET TRADING SIMULATION

The stock simulation strategy proposed by Ding et al. (2015) to evaluate the performance is used of the CAFF method, which simulates the behavior of a daily trader who uses our model in a simple way. CAFF is compared with the best-performing CapTE to illustrate the potential value of our method. The detailed simulation strategy is implemented as follows: if the model indicates that the price of a single stock will rise the next day, the trader will invest $10,000 worth of that stock at the opening price. After the purchase, the length of time the trader holds the stock is determined by the actual market changes. If the stock can earn a profit of 2% or more in the next period of time, the trader immediately sells it; otherwise, traders sell the stock at the closing price at the end of the day. In addition, if the model indicates that the price of a single stock will fall, the same strategy can be used to short the stock. If the trader can buy the stock at a price that is 1% lower than the short-selling price, the trader will buy the stock to cover the position; otherwise, the trader buys the corresponding stock at the closing price.

In Table 2, we selected 12 representative stocks in 22 trading days (30 calendar days) to illustrate the specific profits of the two models, with the maximum return exceeding 20%. The experimental results consistently illustrate that CAFF can obtain higher profits than CapTE and has a superior practical application value. In addition to the reasons analyzed in Section 4.4, we note that if there is no financial data on the stock market the previous day, the two models cannot predict the stock price trend for the day because they have no information for quantitative analysis. Not only will this scenario reduce the accuracy and MCC, it will also harm actual profits. Furthermore, we take the specific market returns at that time as the benchmark, and the returns of the two models are generally higher than the average returns of the actual market. The monthly return of the historical real stock market is 8.35 %, and the average market return obtained by CAFF is 9.3 %.

## 5 CONCLUSION AND FUTURE WORK

We study a popular research topic in current financial market, that is, how to fuse market prices and text information, and quantitatively analyze the key role of source information to enable the predictive model to process different financial data in parallel in an intelligent manner. To solve these problems, we proposed a collaborative attention Transformer fusion model to perform financial market forecasting using historical text and prices. The proposed method is inspired by the cross-modal fusion method and accounts for the characteristics of financial data. According to the quantitative analysis results from the above experiments, text play a key role in the overall framework and are essential to improving the experimental accuracy. Therefore, advanced NLP technology must be used to process text and analyze context relations. How to choose a method that is suitable for analyzing time series information and has practical application value will be addressed in future research.

ETHICS STATEMENT

Studies that doesn't involve human subjects, practices to data set releases, potentially harmful insights, methodologies and applications, pontential conflicts of interest and sponsorship, discrimination/bias/fairness concerns, privacy and security issues, legal compliance, and research integrity issues.

REPRODUCIBILITY

The dataset is available on github:https://github.com/yumoxu/stocknet-dataset.

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
