# OpenReview forum: "A Collaborative Attention Adaptive Network for Financial Market Forecasting"
_ICLR.cc/2022/Conference — ICLR 2022 Submitted_

### Official Review · Reviewer_GNAc · 2021-11-02

**Correctness:** 2
**Technical Novelty And Significance:** 2
**Empirical Novelty And Significance:** 2
**Recommendation:** 3
**Confidence:** 5

**Main Review:**

Comments
This paper proposes an approach to fuse text information and stock prices. However,  I have some concerns regarding the current form of the paper.
1. The technical contribution and novelty are limited as the proposed architecture is composed of several existing techniques.
2. The paper quality is poor; there are many fundamental grammatical errors and mismatches in the paper, some of which are listed below.
- In the abstract, the sentence ``Extensive experimentation is performed on tweets and historical price of stock market, our method can achieve a better accuracy compared with the state-of-the-art methods on two evaluation metrics.'' is not grammatically correct.
- There is a mismatch between figure and paper writings, e.g., the authors mention ConvLSTM in Figure 1, the motivation and the implementation of which are, however not mentioned in this paper.
- Unclear description of the math formulas, e.g., What is the definition of x in (17)? What is the definition of the operator ⊙ in (20)? What do v_i' and v_m' mean in (18) and (19)? What do z_t and z_T mean in (21) and (22)?
- The figures should be vector graphics.
3. Some detailed information is missing, .e.g, there are no details of the used dataset (What is the date range of the datasets? How to split the dataset into training, validation, and test data? [The authors only mentioned they split the data ``by date,'' which is vague.])


**Summary Of The Paper:**

This paper proposes a method to fuse tweets and stock prices for stock trend prediction flexibly. The authors claim that the proposed method outperforms other existing fusing methods. Furthermore, according to the results of the market trading simulation, this method achieves higher profits than other methods.


**Summary Of The Review:**

Overall, the quality of this paper in terms of both technical contributions and presentation does not meet the bar of publication of ICLR. (see the above comments)

---

### Official Review · Reviewer_TVcH · 2021-11-02

**Correctness:** 3
**Technical Novelty And Significance:** 2
**Empirical Novelty And Significance:** 2
**Recommendation:** 5
**Confidence:** 4

**Main Review:**

Strengths:
- This method is well-motivated and the problem it solves is interesting and important.
- It achieves state-of-the-art performance in a stock prediction dataset.

Weaknesses:
- This paper utilizes most of the spaces on Page 3, 4, and 5 to describe some well-established concepts like LSTM, attention, self-attention, and Transformer layer. This leads to several potential problems.
    - It emphasizes too much on some technical details that are not strongly related to the proposed model and contributions (e.g., the difference between GELU, ReLU, and ELU), but ignores many details of their own implementation.
    - On the one hand, it is hard to identify the novelty of the proposed model. It would be very helpful to compare the proposed modules with existing modules such as Transformer. For example, the multi-features fusion module looks like a Transformer Encoder Layer. If it is true, it would be much better to specifically claim that "we use a Transformer Encoder Layer for feature fusion, where the query is xxx, the key is xxx" instead of renaming it as "multi-features fusion" and describe each component of it in detail. This may lead to a lot of confusion.
    - On the other hand, it is hard to understand the technical details of the proposed model. For example, this paper describes the formula of multi-head self-attention but does not give the details of how they construct query/value/key and what the dimension of each vector is, while the first part is well-known and the second part is what this paper really propose.
- Some minor problems/suggestions:
    - Figure 1 and Table 2 are out of the boundary
    - I think it would be very helpful to align the name of the sections (e.g., Sec 3.1.2) with the module names in Figure 1 (e.g., multi-features fusion). This will make it much easier for readers to align your text description with the figure.

Questions:
- What is a training sample (e.g., an (x, y) pair) of the proposed model? Based on the description, the output (y) is a binary label that indicates the rise/down of a specific stock. What is the input (x)? Is it the combination of all the tweets about a specific company on a specific date together with the prices of the company on that date?
- Are you using the price vectors as value/key and using the text vector as the query?
- What does 07/06/2021 mean? Based on the context, I guess it means 2021 June 7th?

**Summary Of The Paper:**

This paper proposes a co-attention-based model that fusion the representation of tweet and stock price data to make stock trend predictions. The proposed method achieves state-of-the-art performance in a stock prediction dataset.

**Summary Of The Review:**

This paper solves the interesting text-based stock prediction problem with a well-motivated method. However, it is hard to identify the novelty and understand the method in detail based on the current manuscripts. I think this paper can be greatly improved when more details and comparisons with existing methods are provided.

---

### Official Review · Reviewer_K1Z9 · 2021-11-04

**Correctness:** 3
**Technical Novelty And Significance:** 3
**Empirical Novelty And Significance:** 2
**Recommendation:** 5
**Confidence:** 3

**Main Review:**

**Strength**
1. It's a meaningful idea to process text and stock prices in parallel and then fuse them.
2. Using a co-attention transformer to fuse text and stock information is a novel idea that has not been explored.

**Weakness**
1.  The writings of some parts need more polish
    * Some references are not in the right places. For example, no reference of the dataset is provided when the dataset is introduced in Sec. 4.1 and no references about temporal attention mechanism are provided in Sec. 3.1.4.
    * Paragraph 2 of Section 2 can be dived into 2 paragraphs with the second one focusing on fusion technology.
    * Section 3.1 is not so clear to me. What does endpoint mean? Is it the last day of input data? Does 03/07/2021 refer to March 7th, 2021? What this date and 07/06/2021 the endpoints for 08/06/2021?
2. It's meaningful to use Bi-LSTM and self-attention to model the text. But why not take one step further to apply Transformer, which is a more powerful model, to encode the text?
3. The intermediate results of the proposed model are not very clear, e.g. what are the input and output dimensions of each component?
4. The motivations of some model components are not well explained. Why an LSTM is needed after the co-attention transformer, is it used to aggregate the feature sequence? Why is temporal attention needed afterward?
5. More details are needed for the market trading simulation experiments for other researchers to replicate it, eg. what's the time period used in it, why are these 12 stocks selected, can we apply this simulation on all stocks, and get some quantitative results?
6. Does the stock price input data only go through a normalization? If so, it looks the model just directly uses the raw stock price data instead of doing the claimed parallel processing. Besides, what's the dimension of the input and output? More model details would be very helpful.


**Summary Of The Paper:**

This paper proposed a novel approach to jointly model text and stock price information and fuse them for stock market forecasting. It encodes text and stock price information in parallel and then fuses them using a co-attention transformer. Empirical results over a real-world dataset and trading simulations demonstrate that the proposed approach can outperform the existing baselines.

**Summary Of The Review:**

This paper proposed a novel approach to jointly model text and stock price information and fuse them for stock market forecasting. While the proposed approach is solid, this paper still needs more work on its writing and experiments.

---

### Decision · Program_Chairs · 2022-01-20

**Decision:**

Reject

**Comment:**

This paper proposed a new approach to jointly model text and stock price information and fuse them for stock market forecasting. It encodes text and stock price information in parallel and then fuses them using a co-attention transformer. According to the reviewers, the design of the model is not very well justified and seems to be a little ad hoc. The authors spent quite a few pages introducing background knowledge and the novelty of the proposed model is not sufficiently described. Some details in the experiments are missing, and it is not clear whether the results could be easily reproduced.  There are many writing issues too. As a result, we do not think the paper is ready for publication at ICLR in its current form. BTW, after the reviewers posted their comments, the authors did not submit their rebuttals.